# An experimental test of the effects of redacting grant applicant identifiers on peer review outcomes

**Richard K Nakamura[1], Lee S Mann[1], Mark D Lindner[2], Jeremy Braithwaite[3†], Mei-Ching Chen[2], Adrian Vancea[2], Noni Byrnes[2], Valerie Durrant[2], Bruce Reed[2]\***

[1]Retired, formerly Center for Scientific Review, National Institutes of Health, Bethesda, United States; [2]Center for Scientific Review, National Institutes of Health, Bethesda, United States; [3]Social Solutions International, Rockville, United States

## Abstract

**Background:** Blinding reviewers to applicant identity has been proposed to reduce bias in peer review.

**Methods:** This experimental test used 1200 NIH grant applications, 400 from Black investigators, 400 matched applications from White investigators, and 400 randomly selected applications from White investigators. Applications were reviewed by mail in standard and redacted formats.

**Results:** Redaction reduced, but did not eliminate, reviewers' ability to correctly guess features of identity. The primary, preregistered analysis hypothesized a differential effect of redaction according to investigator race in the matched applications. A set of secondary analyses (not preregistered) used the randomly selected applications from White scientists and tested the same interaction. Both analyses revealed similar effects: Standard format applications from White investigators scored better than those from Black investigators. Redaction cut the size of the difference by about half (e.g. from a Cohen's $d$ of 0.20–0.10 in matched applications); redaction caused applications from White scientists to score worse but had no effect on scores for Black applications.

**Conclusions:** Grant-writing considerations and halo effects are discussed as competing explanations for this pattern. The findings support further evaluation of peer review models that diminish the influence of applicant identity.

**Funding:** Funding was provided by the NIH.

**\*For correspondence:** bruce.reed@nih.gov

**Present address:** †EvaluACT, Inc, Playa Vista, United States

## Introduction

National Institutes of Health (NIH) distributes over $34 billion per year in research grants to support biomedical research at research institutions and small businesses across the United States. NIH funding is important, not only to the future of scientific discovery, but to the careers of individual scientists. Grant funding enables them to pursue their scientific studies and success in obtaining NIH funding often factors into tenure and promotion decisions. In 2011, the National Academies of Science (NAS) issued a report arguing that increased diversity in the scientific workforce is critical to ensure that the United States maintains its global leadership and competitive edge in science and technology (*National Academy of Sciences, N.A.o.E, 2011*). The same year, Ginther et al. reported that the likelihood of Black PIs being awarded NIH research funding between 2000 and 2006 was 55% of that of White investigators (*Ginther et al., 2011*). This funding gap persists (*Ginther et al., 2011*; *Ginther et al., 2018*; *Hoppe et al., 2019*; *Erosheva et al., 2020*) and the proportion of NIH research grant applicants who are Black has increased only slightly in the ensuing years, from 1.4% in Ginther's data to 2.3% in 2020.

The largest single factor determining the probability of funding in the highly competitive NIH system is the outcome of peer review. Peer review panels meet to discuss and score applications according to scientific merit; the possibility that peer review is biased is of great concern to applicants, funding agencies, and the American public (*Fox and Paine, 2019*; *Gropp et al., 2017*; *Hengel, 2017*; *Lerback and Hanson, 2017*; *Wennerås and Wold, 1997*; *Taffe and Gilpin, 2021*). Disparities in success rates of scientific publishing (*Bendels et al., 2018*; *Hopkins et al., 2012*; *Ouyang et al., 2018*) and in obtaining grant awards (*Ginther et al., 2011*; *Pohlhaus et al., 2011*; *Ginther et al., 2016*) raise questions of whether reviewer bias on applicant demographics (race, gender, age, career stage, etc.) or institutional reputation unfairly influence the review outcomes (*Wahls, 2019*; *Witteman et al., 2019*; *Murray et al., 2016*). Concerns are particularly salient for NIH because the criteria for evaluating the scientific merit of research projects include 'investigators' and 'environment', thus explicitly directing reviewers to take these factors into account.

It should be understood that NIH funding is not determined by peer review alone, but rather is additionally determined by scientific priorities and budgets at the funding institutes. Funding rates for major research grants vary approximately threefold, from about (10% to nearly 30%) across the institutes meaning that applications in some areas of science enjoy greater success than others. A recent paper focused attention on the finding that funding success varied substantially depending on scientific topic, and that the topics most often studied by Black investigators tend to have low funding rates (*Hoppe et al., 2019*). An important follow-up paper showed that this association was primarily attributable to the disparate funding rates across the 24 NIH institutes, rather than topical bias in peer review (*Lauer et al., 2021*). Nonetheless, peer review outcomes are a fundamental determinant of success across the NIH.

Various approaches to reducing demographic bias in review have been proposed. Blinding reviewers (in 'double blind' or 'dual anonymous' reviews) to applicant identity and institutional affiliations is one such approach (*Cox and Montgomerie, 2019*; *Fisher et al., 1994*; *Haffar et al., 2019*; *Okike et al., 2016*; *Snodgrass, 2006*). The literature examining the impact of blinding on review outcomes is mixed. With respect to gender, for example, blinding has been reported to reduce disparities (*Budden et al., 2008*; *Terrell et al., 2017*; *Aloisi and Reid, 2019*) but has also been ineffectual (*Primack, 2009*; *Whittaker, 2008*; *Blank, 1991*; *Ledin et al., 2007*). To our knowledge, there are no studies evaluating real review of scientific grants blinded with respect to race. Even so, the importance of diversifying science, the strong correlation between review and funding outcomes, and the perceived tractability of review make review interventions attractive. Strong concerns about the potential of demographic bias, and especially anti-Black racial bias, in peer review remain (*Stevens et al., 2021*).

The present study was part of the NIH response to Ginther's 2011 report of the Black-White funding gap and the NAS report (*National Academy of Sciences, N.A.o.E, 2011*) on the lack of diversity in the U.S. scientific workforce. An NIH Advisory Committee to the Director (ACD) Working Group on Diversity in the Biomedical Research Workforce (WGDBRW) recommended that 'NIH should design an experiment to determine the effects of anonymizing applications…' in peer review (*Working Group on Diversity in the Biomedical Research Workforce (WGDBRW) and T.A.C.t.t.D. (ACD), 2012*).

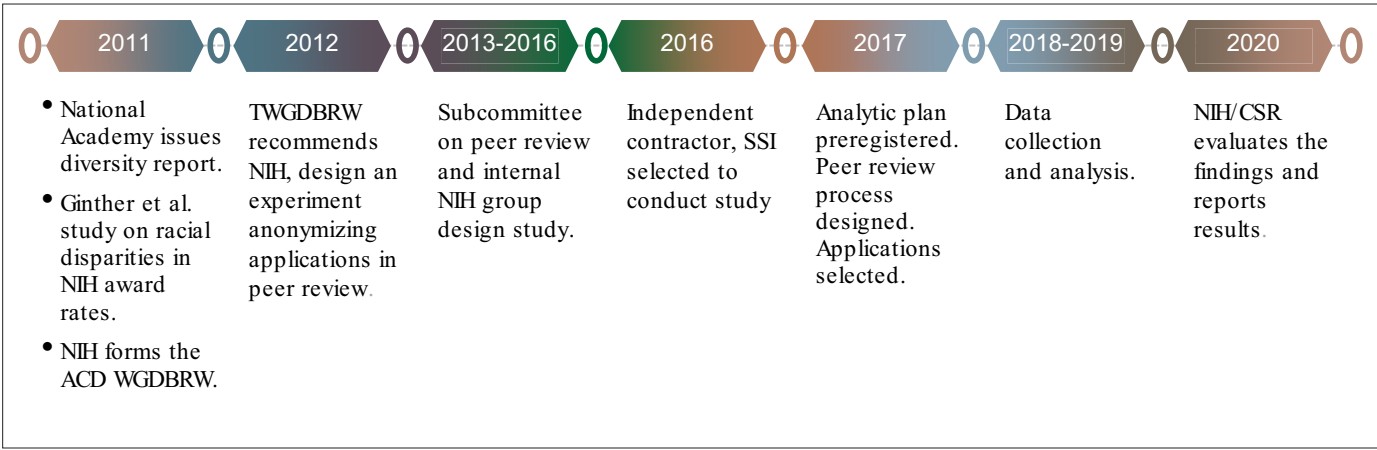

**Figure 1.** Study background and timeline.

An NIH Subcommittee on Peer Review (*National Institutes of Health, 2013*) and an internal NIH group led by then-CSR Director (RN) designed it as a large-sample experimental investigation of the potential effects of racial bias on review outcomes, specifically bias against Black or African American investigators (see *Figure 1*).

The decision to restrict the focus to Black scientists stemmed from three considerations. First, although funding gaps have been reported for other disadvantaged groups, none has been persistently as large as that experienced by Blacks. For example, in 2013 NIH award rates for major research projects for Hispanic scientists were 81% that of White investigators, and award rates for Asians 83% that of Whites. Both rates improved to 88% that of Whites in 2020. Past differences in funding rates for male versus female applicants for major NIH research awards have disappeared (https://report.nih.gov/nihdatabook/report/131). Second, the funding disparity for Blacks occurred in the context of the U.S. history of centuries long, pernicious and pleiotropic anti-Black racism. Lastly, the scale of a properly powered experiment to investigate multiple forms of demographic bias was prohibitive from a practical perspective.

The experimental intervention was to anonymize applications by post-submission redaction of applicant identity and institutional affiliation. Using real, previously reviewed NIH R01 applications, the experiment compared scores for applications from Black vs. White applicants as re-reviewed for this study in their standard (original) vs. redacted formats. The primary research question was 'Does concealing the race and identity of the applicant affect reviewers' overall impact scores of applications from Black and White applicants differently?'.

## Materials and methods

### Design

The study was conducted by a contract organization, Social Solutions International (SSI). The study design and analytic plan were preregistered at the Center for Open Science in October 2017 (https://osf.io/3vmfz). The experiment obtained reviews using either the standard (original) application format or the redacted format for applications from Black PIs vs. White PIs. Applications were real NIH R01s (NIH's major research project awards) that had been previously reviewed in 2014–2015 by CSR. There were three sets of applications; 400 R01 applications submitted by Black PIs, and two comparator sets from White PIs, one selected to match the Black PIs' applications on review-relevant features, the other selected randomly. All applications were redacted to obscure the PI's identity, race, and institutional affiliation. The original and redacted versions of each application were re-reviewed independently for this study by new reviewers. Each reviewer provided for each application (1) a preliminary overall impact score (which is the primary outcome measure), (2) a written critique, (3) guesses of the race, ethnicity, gender, institutional affiliation, career stage, and name of the investigator, along with confidence ratings regarding those guesses, and (4) ratings of grantsmanship. Grantsmanship was measured with two items intended to measure aspects of the application evaluation *not* captured by the overall impact score or five individual criterion scores: (1) Is the application organized, well-written, and easy to follow? ('Grant 1'), and (2) Did the application enable the review to generate informed conclusions about the proposed project? ('Grant 2'). The major hypothesis was that redaction would differentially affect the scores given to Black and White application sets; that is, either Blacks would score better, Whites worse, or both when applications were redacted.

### Sample

The preregistered plan called for a sample size of 400 per group based on power calculations for a two-sample t-test with alpha = 0.05, an effect size of 0.25, and power of 94%. As documented in the Transparent Change Summary (available on request), linear mixed models were used instead of the originally registered t-tests and the central hypothesis was tested with an interaction term. A revised power analysis focused on detecting interactions in a mixed-effects linear regression showed that with an N of 400 per cell, the study had 70% power to detect an effect size (*d*) of 0.2, 80% to detect an effect size of 0.25, and greater than 90% power to detect an effect size of 0.3 (*Leon and Heo, 2009*; *Supplementary file 1A*).

The 400 R01 applications from Black contact PIs comprised nearly 80% of such applications that were reviewed in 2014–2015. The plan specified that a sample of 400 applications from White PIs

**Table 1.** PI demographics and application characteristics by sample.

| Match criteria | Black (n = 400) | White matched (n = 400) | White random (n = 400) |
|---|---|---|---|
| **Gender** | | | |
| Male | 232 | 233 | 276 |
| Female | 166 | 167 | 120 |
| Unknown | 2 | | 4 |
| **Institution NIH mean (SD) awarded dollars in $millions** | 182.88 (172.02) | 171.12 (159.85) | 176.92 (157.13) |
| **Type of application** | | | |
| Type 1 (New) | 370 | 369 | 334 |
| Type 2 (Renewal) | 30 | 31 | 66 |
| **Revision or resubmission** | | | |
| A0 (original submission) | 290 | 290 | 263 |
| A1 (resubmission) | 110 | 110 | 137 |
| **Early stage investigator** | | | |
| Yes | 102 | 102 | 47 |
| No | 298 | 298 | 353 |
| **Investigator age mean (SD)** | 48.66 (9.31) | 50.27 (10.20) | 51.96 (9.96) |
| **Behavioral/social science IRG** | | | |
| Yes | 174 | 173 | 75 |
| No | 226 | 227 | 325 |
| **Degree held** | | | |
| MD | 80 | 72 | 54 |
| PhD | 237 | 267 | 289 |
| MD/PhD | 37 | 33 | 40 |
| Others | 24 | 16 | 8 |
| Unknown | 22 | 12 | 9 |
| **Original preliminary overall impact scores: mean (SD)** | 4.35 (1.46) | 4.34 (1.36) | 3.94 (1.26) |
| **% with multiple PIs** | 24 | 18 | 21 |

matched to the Black PI applications on preliminary overall impact score and on review-relevant characteristics would be the comparison group for the primary test of the hypothesis. A secondary comparison set of 400 applications from White PIs, randomly selected from the approximately 26,000 reviewed in 2014–2015 was also drawn ('random White sample'). The random White sample was to provide a 'real world' comparator, and an alternative test of the hypothesis (*Rubin, 2006*; *Campbell and Stanley, 1963*).

NIH applications may have more than one principal investigator, for example, may be a 'multiple PI' (MPI) application. MPI applications were assigned to groups based on the demographics of only the contact PI. Overall, 21% of applications were MPI. See *Table 1* for sample characteristics.

## Matching and redaction

The intent of using matched sets of applications from Black and White PIs was to isolate the effect of PI race and redaction on review outcomes. Applications were matched on actual preliminary overall impact scores and seven additional variables: (1) area of science (behavioral/social science vs. other), (2) application type (new/renewal), (3) application resubmission or not, (4) gender, (5) early-stage

investigator (ESI) or not, (6) degree of PI (PhD, MD, etc.), and (7) institutional research funding (quintiles of NIH funding) (***Supplementary file 1B***).

Redaction was performed by a team of 25 SSI research staff and checked by a quality assurance team member. Redaction took between 2 and 8 hr per application to accomplish, and quality assurance took 2–4 hr more (redacted fields listed in ***Supplementary file 1C***).

## Review procedures

Reviews were overseen by nine PhD-level scientists contracted by SSI, who functioned as scientific review officers (SROs). Three had retired from CSR, and one had previous contract SRO experience at CSR. The other five had no prior SRO experience with NIH. All SROs were provided with 6 hr of group training along with individual coaching from NIH-experienced SROs. Reviewers were recruited by the SROs from more than 19,000 scientists who had served on the study sections where the 1200 applications were originally reviewed. Reviewers were recruited using a standardized email invitation that stated that this was a study being conducted 'to examine the impact of anonymization on NIH peer review outcomes in support of CSR's mission to ensure that grant review processes are conducive to funding the most promising research'. Reviewers were told nothing about the racial composition of the application sample.

Reviewers were assigned applications based on expertise: SROs reviewed the application project narrative, abstract, specific aims, and research strategy and tagged each application with key words to indicate the topic and methods. SROs then matched applications to potential reviewers' Research, Condition, and Disease Categorization (RCDC) terms and scores. RCDC terms are system-generated tags applied by NIH to all incoming applications, designed to characterize their scientific content. Weighted combinations of scores can be used to characterize the content of an application or to characterize a scientist's expertise.

Six reviewers were recruited for each application; three were randomly assigned to review the standard application, three to review the redacted version. Most reviewers reviewed some standard format and some redacted applications. The goal was for each reviewer to get ~6 applications to review but problems with reviewer recruitment and attrition resulted in 3.4 applications per reviewer on average (median = 3, interquartile range = 1–5, maximum 29).

In standard NIH peer review, each reviewer scores the application on overall level of scientific merit before reading other reviewers' critiques ('preliminary impact score'—the outcome for this study), then may read other's critiques and adjust that score, then presents that preliminary score to the panel, explains the basis for it, the panel discusses the application, reviewers revise their scores, and each panelist votes a final score. This procedure was considered not feasible for this study. Instead, review was done entirely through non-interactive written reviews. Reviewers were given a chart of the NIH scoring system (1 = best, 9 = worst) and standard R01 critique templates. In addition to providing an overall impact score, reviewers rated applications on grantsmanship and on whether redacted applications provided enough information to enable a fair review. Reviewers reviewed each application as a package, beginning with the writing of the critique and scoring of the application, ending with the questions on grantsmanship and guesses about applicant/institutional identity. The review template and additional rating items are in ***Supplementary file 2***. Nearly all applications received the desired six critiques (7155 of 7200).

## Statistical analysis

The preregistered protocol defined three primary questions of interest: (1) Effectiveness of redaction in achieving anonymization: Are reviewers less accurate in their assessment of the applicants' actual race in the anonymized version of the applications? (2) Effectiveness of the matching procedure: Did the matching produce equivalent preliminary overall impact scores in the current study on the standard application format? (3) Primary test of the study hypothesis: Does concealing the race and identity of the applicant affect reviewers' preliminary overall impact scores of applications from Black and White applicants differently?

Question 1 was evaluated using chi-square analyses comparing rates of correct identification of Black and White PIs using standard format and redacted applications. Questions 2 and 3 were examined using linear mixed models (multi-level models) to account for the intra-class correlation of impact scores within individual applications. The average of the three reviewers' preliminary overall impact

**Table 2.** Reviewer's guesses of applicant race in relation to actual race by application format.

| Reviewer guess of PI race | Standard format applications | | Redacted format applications | |
|---|---|---|---|---|
| | Black PIs | White PIs | Black PIs | White PIs |
| Black | 683 (58%) | 49 (2%) | 336 (28%) | 48 (2%) |
| White | 432 (36%) | 2234 (93%) | 723 (61%) | 2081 (87%) |
| Other | 45 (4%) | 66 (3%) | 78 (7%) | 172 (7%) |
| No guess | 25 (2%) | 41 (2%) | 52 (4%) | 90 (4%) |

scores for each application is the dependent variable. The model has two binary main effects, PI race and application format. Thus, the primary test of the study hypothesis is tested with the race × application format interaction term. A significant interaction would indicate that the effect of redaction on scores was different for Black and White applications.

The preregistered plan specified that the hypothesis be tested using the matched samples of applications from Black and White PIs (in order to maximize statistical power), and that the randomly selected set of applications from White investigators would be used for secondary analyses. For clarity of presentation, methods for the secondary models are described with the results of those models.

## Results

### Preregistered question 1

Question 1 concerns the effectiveness of redaction in achieving anonymization. *Table 2* shows redaction reduced the rate at which reviewers could guess PI race for Black PI's by over half, from 58% to 28%. The effect on the rate of correctly guessing the race of White PIs was much smaller (93%–87%). Reviewers mistakenly guessed that Black PIs were White 36% of the time with standard format applications, 61% of the time with redacted applications. (Data for the two White samples were combined for simplicity, because their distributions were highly similar.) Reviewer confidence in their guesses of race using redacted applications was just over 2 on scale from 1 ('not at all confident') to 5 ('fully confident') for all PI samples. Using standard format applications, confidence ratings for guesses of race were about one point better; ratings did not vary appreciably by applicant race (see *Table 3*). Guesses of PI race based on redacted applications were significantly less likely to be correct than were guesses based on standard applications; $\chi^2(1)$ = 160.2, p < 0.001.

Reviewers of redacted applications were asked to guess the PI/research group. Most of the time they did not venture guess, but 21% of the time, a reviewer was able to make an exact identification. *Table 4* details these data according to application set. Guesses for MPI applications were counted as correct if the reviewer named any one of the PIs.

Thus, in answer to question 1, redaction diminished but did not eliminate reviewer knowledge of applicant race and identity. Reviewers were about half as likely to identify applicants as Black when viewing redacted applications compared to standard applications.

**Table 3.** Reviewer confidence regarding their guesses of investigator demographics.

| Applicant characteristic | Black investigators | | White matched investigators | | White random investigators | |
|---|---|---|---|---|---|---|
| | Standard reviews | Anonymized reviews | Standard reviews | Anonymized reviews | Standard reviews | Anonymized reviews |
| Race | 3.2 | 2.1 | 3.2 | 2.2 | 3.4 | 2.2 |
| Gender | 4.3 | 2.3 | 4.4 | 2.3 | 4.5 | 2.3 |
| Institution | 4.2 | 3.2 | 4.3 | 3.3 | 4.4 | 3.3 |
| Career stage | 4.2 | 3.1 | 4.2 | 3.2 | 4.4 | 3.2 |

Note: 5-point scale, 1 = low confidence, 5 = high confidence.

**Table 4.** Rates of reviewer identification of name/research group in redacted applications.

| PI race | Correct | Incorrect | No guess |
|---|---|---|---|
| Overall (3580) | 21.6% (775) | 6.1% (217) | 72.3% (2588) |
| Black (1189) | 18.9% (225) | 5.6% (67) | 75.4% (897) |
| White (matched sample) (1194) | 19.4% (232) | 7.0% (84) | 73.5% (878) |
| White (random sample) (1197) | 26.6% (318) | 5.5% (66) | 67.9% (813) |

## Preregistered question 2

Question 2 asks did the matching produce equivalent preliminary overall impact scores for standard application format applications? Although the applications sets were matched on the preliminary overall impact scores received in NIH review, simple contrasts show that when reviewed for this study, applications from White PIs scored better (M = 3.9 White, 4.1 Black). The effect size was small, d = 0.20. *Figure 2* shows the distributions of average preliminary overall impact scores for Black, White matched, and White random PI applications in standard and redacted formats.

## Preregistered question 3

Question 3 tests the study hypothesis: Does concealing the race and identity of the applicant affect reviewers' preliminary overall impact scores of applications from Black and White applicants differently? *Table 5* summarizes the analysis, which found a significant main effect for both PI race and application format. On average, applications from White PIs received better scores than those from Black PIs. Redacted format applications scored worse than standard format applications. Both effect sizes were small. The prespecified statistical test of the study hypothesis is the race × application format interaction and was not statistically significant (p = 0.17). Removing from the analyses scores for those cases in which the review correctly identified the PI did not appreciably change the parameter estimates or significance levels.

*Table 6* shows the observed data and simple contrasts. Redaction had a significant effect on White PI's applications (scores became worse). Redaction had no effect on scores for Black PI's applications.

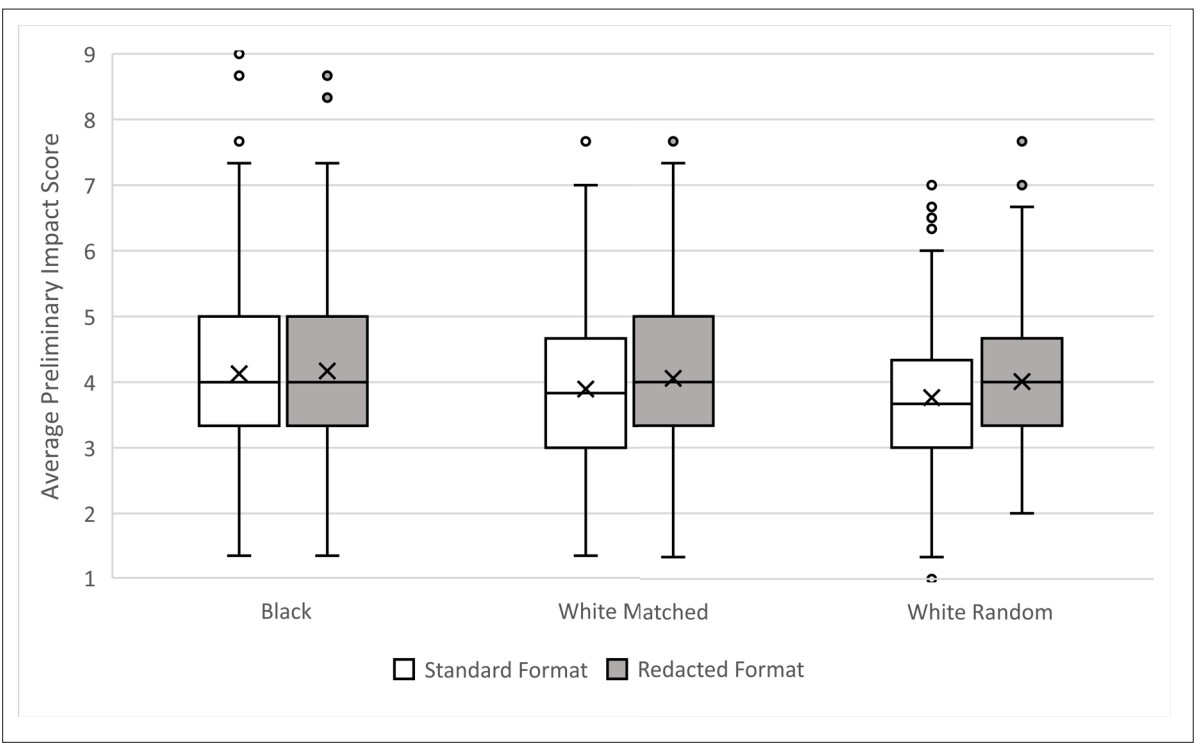

**Figure 2.** Distributions of preliminary overall impact scores according to race of PI and format in which the applications were reviewed. Boxes delineate the central 50% of scores those falling between the 25th and 75th percentiles (Interquartile Range, IQR). Whiskers extend 1.5X the IQR. Dots mark outliers. Horizontal lines within boxes indicate the median, and "x" marks the mean value. Lower scores are better.

**Table 5.** Primary analysis.
Effects of race and application format on overall impact scores in matched White and Black application sets.

| | Estimate | p-Value | 95% Confidence interval (CI) |
|---|---|---|---|
| Fixed effects | | | |
| Race | –0.17 | 0.01 | (–0.31,–0.04) |
| Application format | –0.10 | 0.02 | (–0.19,–0.02) |
| Race × application format | –0.12 | 0.17 | (–0.29, 0.05) |
| Intercept | 4.06 | < 0.001 | (3.99, 4.13) |
| Random effects | | | |
| Application intercept | 0.61 | – | (0.51, 0.72) |

Note: The reference category for race is the Black group. The reference category for application format is the redacted format.

Distributions of change scores (three-reviewer average score, redacted format minus standard format score) for the two samples were similar: for Black and matched White samples respectively the means were 0.04, 0.16; medians 0, 0; 1st quartile –0.67, –0.67; 3rd quartile 1, 1.

## Secondary analyses

Using the White random application set as a comparator provided a secondary test of the study hypothesis in applications representative of those received at NIH (from Black or White PIs).

It also allowed exploratory analyses of additional factors that may influence review outcomes. The dependent variable was the preliminary overall impact score entered by each reviewer. Cases with missing data were deleted. Covariates of interest were categorized as follows: investigator demographics, application characteristics, reviewer perceptions, and grantsmanship indicators. Effects of covariates on final overall impact scores were tested in a set of linear mixed models. The base model included race of the PI as the only predictor. Models 2–4 add blocks of covariates. For each model, appropriate random effects were specified. To determine which random effects were appropriate, we began by including random slopes for all predictors in the model, then used backward list deletion to determine which random effects significantly contributed to the given model. *Table 7* displays the fixed and random effects for the nested models.

Model 1 tested the unadjusted effect of race of the PI on overall impact scores across both application formats. Applications from White PIs scored better. The effect was small, explaining less than 2% of variance in overall impact scores.

Model 2 added application characteristics and additional characteristics of the PI. All covariates except PI gender had significant effects; resubmissions and competing renewals scored better while applications from ESIs and institutions in the lowest quintile of NIH grant funding scored worse. Including these effects reduced the effect of PI race by half, but PI race remained a significant predictor.

**Table 6.** Simple contrasts of average preliminary impact scores for redacted vs. standard format applications by PI race.
Matched White application set.

| Race | Anonymization condition | | Simple contrast (SE) | Effect size |
|---|---|---|---|---|
| | Standard | Anonymized | | |
| Black | 4.13 | 4.17 | 0.04 (0.06) | 0.04 |
| White matched | 3.89 | 4.05 | 0.16* (0.06) | 0.14 |
| Simple contrast (SE) | –0.23* (0.08) | –0.12 (0.08) | | |
| Effect size for race | 0.20 | 0.10 | | |

*p <.05 (Bonferroni-adjusted).

**Table 7.** Parameter estimates and standard errors from nested models predicting overall impact scores in the Black and random White application sets.

| Fixed effects | Model 1 (n = **4764** 800 applications) | | Model 2 (n = **4728** 794 applications) | | Model 3 (n = **4728** 794 applications) | | Model 4 (n = **4315** 794 applications) | |
|---|---|---|---|---|---|---|---|---|
| | Coef. | SE | Coef. | SE | Coef. | SE | Coef. | SE |
| **Demographics** | | | | | | | | |
| Race (White = 1) | –0.266[a] | 0.069 | –0.132[c] | 0.065 | –0.132[c] | 0.065 | –0.124 | 0.068 |
| Type 2 application | | | –0.492[a] | 0.101 | –0.491[a] | 0.101 | –0.484[a] | 0.104 |
| A1 application | | | –0.420[a] | 0.069 | –0.420[a] | 0.069 | –0.415[a] | 0.072 |
| Gender | | | –0.005 | 0.067 | –0.005 | 0.067 | 0.013 | 0.069 |
| Early-stage investigator | | | 0.178[c] | 0.084 | 0.178[c] | 0.084 | 0.186[c] | 0.087 |
| Low NIH institutional funding | | | 0.618a | 0.094 | 0.618a | 0.094 | 0.612[a] | 0.097 |
| **Experimental covariates** | | | | | | | | |
| Format (standard = 1) | | | | | –0.144[a] | 0.042 | –0.022 | 0.041 |
| Format × race | | | | | –0.186[b] | 0.083 | –0.237[b] | 0.080 |
| **Perceptions** | | | | | | | | |
| PI race guess Black | | | | | | | –0.155[b] | 0.069 |
| PI gender guess female | | | | | | | –0.069 | 0.061 |
| PI career stage guess Early-stage investigator | | | | | | | 0.091 | 0.063 |
| Institutional funding guess 'low' | | | | | | | 0.447[a] | 0.134 |
| **Grantsmanship indicators** | | | | | | | | |
| Grant 1 | | | | | | | –0.519[a] | 0.027 |
| Grant 2 | | | | | | | –0.204[a] | 0.029 |
| **Random effects** | | | | | | | | |
| Grant 1 slope | | | | | | | 0.052 | |
| Institution slope | | | 0.489 | | 0.489 | | 0.477 | |
| Application intercept | 0.614 | | 0.400 | | 0.402 | | 0.511 | |
| Residual | 2.044 | | 2.041 | | 2.032 | | 1.561 | |

Note: Statistically significant parameter estimates are bolded; [a]p ≤ 0.001, [b]p ≤ 0.025, [c]p < 0.05.

Model 3 provides a secondary test of the study hypothesis by adding terms for application format and the PI race by format interaction. Application format was significant, with redacted applications scoring worse, and the application format × race interaction was significant. Redaction did not significantly change scores for Black PIs but significantly worsened scores for White PIs. *Table 8* shows the effects of PI race and application format in the raw (unadjusted) data.

Model 4 added reviewer guesses of applicant race, gender, ESI status, institutional funding, and ratings of grantsmanship. Reviewer guesses that the PI was an ESI or was from an institution with low NIH funding were both associated with worse scores, institutional status having the larger effect. Controlling for all other variables in the model, including actual PI race, reviewer's guess that the PI was Black was associated with slightly better scores. Better ratings on grantsmanship indicators were associated with better overall impact scores. In the final model the following indicators were associated with better scores: competitive renewal, resubmission, reviewer ratings of better grantsmanship, and reviewer guess that the PI was Black. The following indicators were associated with worse scores: ESI, low funded institution, and reviewer guess that the institution was in the low funded group. With this set of covariates, neither PI race nor application format was significant, but the interaction of

**Table 8.** Simple contrasts of average preliminary impact scores for redacted vs. standard format applications by PI race.
Randomly selected White application set.

| Race | Anonymization condition | | Difference (SE) | Effect size |
| --- | --- | --- | --- | --- |
| | **Standard** | **Anonymized** | | |
| Black | 4.13 | 4.17 | 0.04 (0.06) | 0.04 |
| White random | 3.76 | 4.01 | 0.25* (0.06) | 0.21 |
| Difference (SE) | –0.37* (0.08) | –0.16 (0.08) | | |
| Effect size for race | 0.31 | 0.15 | | |

*p < .05 (Bonferroni-adjusted).

format by PI race interaction was significant. PI gender was not a significant predictor of scores in any model.

## Discussion

Designed as a test of whether blinded review reduces Black-White disparities in peer review (**Ginther et al., 2011**; **Hoppe et al., 2019**; **Erosheva et al., 2020**), the data are also pertinent to understanding the basis of the advantage that applications from White PIs enjoy in review. The experimental intervention, post-submission administrative redaction of identifying information, reduced but did not entirely eliminate reviewer knowledge of applicant identity. Applications submitted by Black PIs were compared to two samples of applications from White PIs, one matched on review-relevant variables, the other selected randomly. The preregistered analysis defined the primary question to be whether redaction differentially affected scores according to race and specified that it be tested using the matched White set of applications. That interaction term was statistically nonsignificant. A secondary test, using the randomly selected set of White applications and a different modeling approach, was statistically significant. We suggest that it is more useful to focus on the concordant patterns of observed data and overall similarity of the modeled results in the two analyses, rather than on small differences in significance levels.

The following effects were consistent across both samples and both modeling approaches: (1) applications from White PIs scored better than those from Black PIs; (2) standard format applications scored better than redacted; (3) redaction produced worse scores for applications from White PIs but did not change scores for applications from Black PIs. In both the primary and secondary comparisons, redaction reduced the difference in mean scores of Black and White application sets by about half (**Table 6**, **Table 8**). Thus, the data suggest that redaction, on average, does not improve review outcomes for Black PIs but does reduce the advantage that applications from White PIs have in review of standard format applications. Why?

Applications from White PIs tended to score better than applications from Black PIs. This was unexpected when comparing the matched application sets because the samples were closely matched on the scores they had received in actual NIH review. However, it was not surprising that applications from Black PIs scored worse, on average, than randomly selected applications from White PIs. A persistent gap on the order of 50% between award rates for NIH R01 grants to Black versus White PIs has been previously reported (e.g., **Ginther et al., 2011**; **Hoppe et al., 2019**; **Erosheva et al., 2020**), and the correlation between overall impact scores and probability of funding, across NIH, is the same for Black and White investigators (**Hoppe et al., 2019**). The secondary models identified several factors that partially account for the racial difference in scores. Competitive renewals of previously funded projects and resubmissions of previously reviewed applications scored better. These effects are well known and tend to favor applications from White PIs (because the representation of White PIs among established investigators is higher). Conversely, ESI status and being associated with an institution that is at the lowest end of the NIH funding distribution were both associated with worse review outcomes. Together these factors tend to disadvantage Black PIs but do not entirely account for the gap in scores.

Other studies have identified additional contributors to differential racial outcomes, including cumulative disadvantage (*Ginther et al., 2011*), and differences in publication citation metrics (*Ginther et al., 2018*). Citation metrics are associated with many factors (*Tahamtan et al., 2016*), some of which are not linked to the quality of the paper, factors such as differences in publication practices between areas of research (*Piro et al., 2013*), scientific networks (*Li et al., 2019*), coauthors' reputations (*Petersen et al., 2014*), the Matthew effect (*Wang, 2014*), and race of the authors (*Ginther et al., 2018*).

The data reveal little evidence of systematic bias based on knowledge of, or impressions of PI race per se. NIH applications do not have fields for PI race (or gender). In this study reviewers were asked to guess PI race, after reading and scoring the application, not before, and reviewers were generally not very confident of their guesses (see *Table 3*). Thus, how much 'knowledge' reviewers had about PI race, and at what point in the process they formed their impression of race is unclear and likely varies across applications and reviewers. With standard applications, reviewers were more likely to guess that Black PIs were Black than White (58% vs. 36%), but with redacted application more likely to guess they were White than Black (61% vs. 28%); even so, redaction did not change scores for Black PIs. Conversely, redaction did not change the frequency of reviewer guesses that White PIs were White (93% standard, 87% redacted), but redaction did change scores for applications from White PIs. A reviewer's guess that the PI was Black had a very small effect on scores (improving them), controlling for multiple other factors including actual PI race. Interpretation of this effect is statistically and substantively complex. It does not necessarily represent positive racial bias toward Black applicants. Reviewers had reason to presume that they were participating in a study examining the effects of PI race on review outcomes; in this context some might have tried to avoid appearing prejudiced and thus scored PIs they believed to be Black more favorably. It could also be that reviewers judged the science to be better because the PI was perceived to be Black in certain scientific contexts, for example, for an implementation study that hinged on engaging minority communities.

Redacted applications scored worse on average. This is not surprising given that redaction was done administratively, post-submission. The application the reviewers read was not the one written and information lost in redaction may have been important. Retrospectively redacting applications does not simply remove information but also changes the context of the information that remains. Applicants wrote their applications believing that reviewers would be given their names and institutional affiliations and that the other information in the application would be judged in that context. They also, presumably, took into account the fact that NIH grant review criteria include 'investigators' and 'environment', and that these criteria are supposed to be factored into reviewers' final scores. Approximately 28% of reviewers of anonymized applications disagreed with the statement 'I believe that reviewers can provide a fair, thorough, and competent review if applications are anonymized'. Alternatively, or in addition, redacted applications might have done worse because they reduced halo effects (*Kaatz et al., 2014*; *Crane, 1967*) that had inflated the scores of standard applications. Halo effects refer to the tendency to rate something better based on a general good impression of the applicant as opposed to specific relevant information; for example, scoring on the basis of positive reputation rather than the application itself. Absent halo effects, the applications may have scored worse when judged on their merits.

Thus, we believe there are two plausible explanations for why applications from White PIs did worse when redacted. One is that redaction reduced positive halo effects of PI and institution. The interconnected factors of PI reputation, scientific networks, scientific pedigree, and institutional prestige certainly can influence review. They are deeply intertwined with race and tend to favor White PIs over others. If redaction reduced halo effects, it would suggest that blinded review models might improve fairness (*Ross et al., 2006*; *Nielsen et al., 2021*). On the other hand, it may have been that when PI identity was deleted, the scientific narrative lost context and was consequently degraded. This would presumably predominantly affect more senior, established PIs, who are disproportionately White. If the effect of redaction represents a mismatch between writing and review conditions, blinding would not likely have a lasting effect because scientists will adjust their grant writing to conform to new review conditions.

There are practical problems with administrative redaction as an anonymization strategy. Each application took 2–8 hr to redact and quality assurance an additional 2–4 hr. Despite careful removal

of names and other information, redaction was only partially successful in blinding reviewers to PI characteristics. Reviewers of redacted applications correctly identified PI race 70% of the time overall and were able to name the PI or research group in 22% of cases. This result, which is consistent with prior attempts at redaction, suggests that it is not possible to administratively obscure identity in all cases.

What accounts for the unexpected finding that the 'matched' samples of applications from Black and White PIs (selected to match on overall impact scores from their actual NIH reviews) scored differently when those same applications were reviewed as part of this study? We think it likely that the change in White matched scores represents regression to the mean, a problem that is more likely when, as is true here, the groups differ on multiple factors that differ in the same direction (*Campbell and Stanley, 1963*). The set of applications from Black PIs represented almost the entire population of such applications. The matched applications from White PIs were drawn from a population of 26,000 applications which, on average, scored better than the Black applications. Each observed score has a true score component and an error component which is presumed to be random. Selecting applications to match a score worse than the population mean risks selecting applications where the error terms are not random but rather are skewed negatively (making the observed scores worse). When the selected applications were re-reviewed, the error terms would be expected to be random rather than predominantly negative, and thus the mean of the observed scores would improve.

Another possibility is that differences in the conduct of real and study reviews account for the difference. The SROs for the experiment included several naïve to NIH review and the procedures for characterizing reviewer expertise and matching reviewers to applications were much simplified compared to the general practices of CSR SROs. Study reviewers saw many fewer applications than is typical for study section reviewers (~3 for most study reviewers, ~8 for CSR study sections). Because of this, any bias that affects the ranking of applications reviewers apply to 'their pile' would not likely be seen in this study. Also, reviewers knew that they were participating in an anonymization study and many likely suspected their scores would be used to assess bias in peer review. A complete listing of the deviations in the study procedures from the actual NIH peer review process are detailed in *Supplementary file 1D*. Despite these differences in review practices, the overall distribution of scores obtained in the experiment closely approximates the distribution of preliminary overall impact scores seen in the actual NIH reviews of these applications.

Designed as a test of whether blinding reviewers to applicant demographics reduces racial disparities in review outcomes, strengths of the study include a large sample (1200 applications, 2116 reviewers, 7155 reviews), the use of real NIH grant applications, and experienced NIH reviewers. Redaction of institutional and individual identity elements from applications changed scores for White PIs applications for the worse, but did not, on average, affect scores of Black PIs' applications. Although the effect was statistically small, both samples of White PI's applications scored better than the Black PIs applications; in each case redaction reduced the size of that difference by about half. It is possible that redaction highlighted gaps in applications written with the assumption that reviewers would know the PI identity; for example, a field-leader in use of a technique might have intended their name to substitute for methodological details. If that sort of grantsmanship accounts for the redaction effect, implementing partial blinding in review is unlikely to have any lasting benefit. If, however, blinding reduces halo effects and that accounts for the reduction in White advantage, then changing review models could perhaps result in a fairer peer review process.

Post-submission administrative redaction is too labor-intensive to implement on the scale that NIH requires. And, even this careful redaction was quite imperfect in concealing elements of identity. However, there are other methods of blinding reviewers to applicant demographics and institutional affiliations. For example, self-redaction of applications might be more effective, and two stage models of review that require judging the merit of an application's science while blinded to applicant identity are interesting. Development of strategies to ensure a level playing field in the scientific peer review process of scientific grant applications is an urgent need.

## Acknowledgements

We wish to acknowledge the invaluable assistance of the Electronic Records Administration (eRA) team at OER, with particular recognition to Inna Faenson and Aaron Czaplicki. We thank Katrina Pearson and the Division of Statistical Reporting in OER for the power analyses, we thank the support staff at CSR who made this study possible: Amanda Manning, Denise McGarrell, and Charles Dumais, and we thank the SSI contract SROs for dedicated service.

## Additional information

### Competing interests

Richard K Nakamura: now retired, was Director of the NIH Center for Scientific Review (CSR) while the study was designed and implemented.. Lee S Mann: now retired, was employed by CSR.. Mark D Lindner, Valerie Durrant: is employed by NIH/CSR. Jeremy Braithwaite: was employed by the contract research organization that conducted the data collection and initial analysis.. Mei-Ching Chen: MC is employed by NIH/CSR.. Adrian Vancea: is employed by NIH/Center for Scientific Review.. Noni Byrnes: is employed by NIH/Center for Scientific Review. She is the Director of CSR.. Bruce Reed: is employed by NIH, he is the Deputy Director of CSR.

### Funding

| Funder | Grant reference number | Author |
|---|---|---|
| National Institutes of Health | | Richard Nakamura |

Employees of the NIH were involved in study design, in data analysis, data interpretation and manuscript writing. Data were collected, and major data analysis completed, by a contract research organization.

### Author contributions

Richard K Nakamura, Conceptualization, Data curation, Funding acquisition, Project administration, Resources, Writing – review and editing; Lee S Mann, Valerie Durrant, Conceptualization, Writing – review and editing; Mark D Lindner, Conceptualization, Formal analysis, Writing – review and editing; Jeremy Braithwaite, Conceptualization, Data curation, Formal analysis, Writing – review and editing; Mei-Ching Chen, Data curation, Formal analysis, Conceptualization; Adrian Vancea, Formal analysis, Conceptualization; Noni Byrnes, Conceptualization, Resources, Writing – review and editing; Bruce Reed, Conceptualization, Supervision, Writing – original draft

### Author ORCIDs

Mark D Lindner http://orcid.org/0000-0002-8646-2980
Bruce Reed http://orcid.org/0000-0002-1606-8646

### Ethics

Human subjects: All participants gave informed consent to participate in this study in accordance with a protocol that was approved on March 27, 2017 by the Social Solutions, Inc IRB, (FWA 00008632), protocol #47.

### Decision letter and Author response

Decision letter https://doi.org/10.7554/eLife.71368.sa1
Author response https://doi.org/10.7554/eLife.71368.sa2

## Additional files

### Supplementary files

• Supplementary file 1. Additional methodological details. (A) Sample size calculations. (B) Number of matched white applications per matching criteria. (C) Redacted fields and their locations. (D) Differences between standard National Institutes of Health (NIH) review and this study.

• Supplementary file 2. Data collection instrument.

- Transparent reporting form
- Source data 1. Data used in reported analyses.

### Data availability

All data analyzed for the findings presented in this manuscript are included in the supporting files.

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
