## [Decision Letter]

**Acceptance summary:**

The authors, a group scientists and peer review administrators from NIH's Center for Scientific Review (CSR), have attempted to study the effect of redaction of applicant identifiers on review outcome using a selection of grant applications from White and Black investigators. The most remarkable finding was that redaction reduced the score difference between White and Black investigators by half, by affecting the scores of White but not Black investigators. Such unconscious bias, evident on this well crafted study, not only re-emphasizes the need for targeted interventions by the NIH and CSR leadership to prevent such bias, but also reiterates the value of diversification of the reviewer pool. Furthermore, the analysis should be extended to investigators from Latin and Asian descents.

**Decision letter after peer review:**

Thank you for submitting your article "An Experimental Test of the Effects of Redacting Grant Applicant Identifiers on Peer Review Outcomes" for consideration by *eLife*. Your article has been reviewed by 3 peer reviewers, and the evaluation has been overseen by me. The following individual involved in review of your submission has agreed to reveal their identity: Carlos Isales (Reviewer #1).

The reviewers have discussed their reviews with one another, and I have drafted this letter to help you prepare a revised submission.

Essential revisions:

Please respond to Recommendations from all reviewers that will help improve the clarity and veracity of your conclusions. Please also attach a point-to-point response.

*Reviewer #1 (Recommendations for the authors):*

This is a new study but additional information would be helpful:

(1) Little information is provided about the reviewers, what was their characteristics (age, gender, race, degree etc).

(2) My understanding is that the NIH shortened the R01 application from 25 to 12 pages in 2011 with the idea that the reviewers track record (eg. biosketch) would weigh more heavily in the reviewers assessment of their ability to complete the proposed studies. In the current study the biosketch is redacted eliminating that metric. Would they have expected to see less of a difference in scores if the application were 25 pages long?

(3) What was the race breakdown of the MPI's? Did presence of a White PI on an application from a Black PI impact the score?

(4) The authors state that reviewers looked at 3.4 applications/reviewer. However the range was quite broad (1-29). This could introduce bias if some reviewers had large number of reviews. What was the median number of grants reviewed per reviewer?

(5) Established investigators work in specific areas, an experienced reviewer will recognize the topic of the grant as most likely coming from one of these specific investigators. Redaction is not likely to help for this group of investigators. Would caps on number of awards or dollar amount to established investigators help? What percentage of the grants reviewed had PI's with 3 or more NIH awards?

(6) An issue is that the group of reviewers from 2014-2015 is not completely comparable to a reviewer from 2020-2021. Thus, the data presented may really only applies to one see specific point in time in the review process.

(7) As the authors acknowledge the dynamics of scoring and review are very different when the process involves meeting and defending your review in front of a group of your peers. Is there a way to assess whether the "quality" of the reviews was different. For example was a subset of reviews sampled by a third party and assessed as high quality?

*Reviewer #2 (Recommendations for the authors):*

1) Please clarify the areas of expertise for the reviewers versus the areas of applications.

2) Please include a general breakdown of applications from varying races to the NIH.

3) Please include funding rates (%'s) for each race class. This really should be included in the beginning to help the readers understand the overall composition of grants versus race and their funding percentages.

4) For increased clarity about early stages, please spell out ESI in the table legends.

*Reviewer #3 (Recommendations for the authors):*

Introduction: The opening lines jump straight into "the process" of peer review rather abruptly. This paragraph could be re-arranged slightly for a better lead-in. NIH needs to be spelled out and maybe a good entry would be a description of how much in public funds the NIH distributes for research purposes. Then the sentence at lines 43-44 should be put before the launch into bias in peer review part. This would then permit the end of the sentence on Line 38 to include mention of the interests of the US public who pay for this research, for completeness. This is not merely a matter of idle academic fluffery and the individual interests of scientist, and this should not be lost with inside-baseball framing.

Introduction, Line 54-56. There are so many obvious flaws with the work of Forscher et al., that it is inappropriate to cite unquestioned as a negative outcome. They used laughably cartoonish stimuli, did not account for reviewer penetration of their intent and even discovered that many of the reviewers caught on to them. The present study is a considerable improvement and the only place for mentioning Forscher et al., is in drawing a contrast in how this study has improved on their glaring shortcomings.

Introduction, Line 57: Somewhere in here the authors should point out that not all peer review is the same and highlight the degree to which NIH grant review focuses explicitly, as one of five co-equal criteria, on the Investigator. This is one of the contributing reasons why available prior work may not be sufficient and further underlines the importance of their investigation. There are important differences between NIH grant review and, say, evaluation of legal summaries or resume-based job call-backs, familiar to some of the most frequently mentioned literature on this topic. This should be revisited in the Discussion around the comments being made in Line 338-341.

Introduction, Lines 60-64: This should be considered as an alternate opening to the Introduction. Much catchier than either the comparatively narrow academic issue of peer review, or even the suggestion made above to open more broadly with the mission of the NIH.

Introduction: While it is understood why the NIH has focused so exclusively on the disparity for Black investigators after the 2011 report, there should be some explicit recognition in the Introduction that the original Ginther found various disparities for Hispanic and Asian PIs that have not been as rigorously examined in follow-up studies such as the present one.

Design: The description of reviewer tasks needs to better clarify the procedure. Were reviewers asked to provide their scores and only afterward asked all of the probing questions likely to tip them off about the true purpose of the study? Or did they know before doing their scoring that they would be asked about whether they guessed PI demographics, who it was and ratings of grant writing/preparation issues.

Design: It is disappointing in the extreme that five of the nine SROs recruited for the study had no prior experience as an SRO. SRO reviewer selection and assignment is a hugely influential and almost entirely opaque aspect of grant review. It would seem essential in a study of this nature to include only experienced SROs. Obviously this cannot be fixed but it should be mentioned as a significant caveat in the Discussion. The points being made at lines 297-298 would be a good place- one possible reason for the unexpected outcome is that the experimental reviewers were somehow systematically different from real panel members.

Review Procedure: The range of reviews per-reviewer was 1 to 29. It would be useful to provide better summary statistics, i.e., include the median and Inter-quartile range for this. It is an important issue and it may be best to trim the sample a bit so that the loads are more balanced and more reflective of typical reviewer loads. These authors know perfectly well that real CSR reviewers have pressures that lead to ranking within their piles, putting closely-meritorious proposals in micro-competition. This is very likely a place where very small and implicit biases could act and this should definitely be addressed in the Discussion, the paragraph from Line 369-381 would be a good place. But this study/data set would appear to be a key opportunity to evaluate if, for example, mean scores of reviewers that only have 1-3 apps to review differ from those from reviewers who have 8-10 to review (a more typical load).

The results of Pre-registered Question 1 describe the reviewer confidence on guessing race for redacted applications as "low" and to be "modestly higher" for standard applications. Yet the rather dramatic and study-relevant decrease in of identification of Black applicants from 59% to 30% (highly relevant when success rates overall at the NIH are only about 11% for Black applicants, going by the published data) is not characterized in such interpretive terms. Suggest being consistent in Results terminology. Either leave all such shading to the Discussion or include accurate shading for all Results sections.

One important addition to the study under Q1 would be to assess scores for misses and false alarms relative to accurate identifications of PI race. This would depend on there being applications on which some reviewers inaccurately identified race and other reviewers guessed correctly, but this should be summarized in any case. It would seemingly be important if, for a given application, reviewers were either all likely to mis/identify or were more randomly distributed.

The results of Pre-registered Question 2 describe the scoring distributions for the three groups under both standard and blinded conditions. They note the difference between standard scores for the Black and white "matched" groups despite being matched on original scores from the initial review process. This outcome indicates perhaps that there is inconsistency in the outcome of review based on race, which would seem to be critical for the NIH to understand better. Although any given application is reviewed once, given amendments and the need for multiple similar proposals to be submitted it is relevant if an applicants' race dictates higher or lower variability in scoring. It is critical to include two additional plots. First, a correlation, by group, between the original scores and the ones generated in this test under each of the standard and blinded conditions. Second, the authors should include a repeated-measures type plot where the individual applications' standard and redacted scores within this study can be visualized. This may require breaking the 400 proposals into score range bins for clarity but it would be crucial contribution to understanding what is driving effects.

The results of Pre-registered Question 3 address the study's main hypothesis. This is the main point and the data should be provided in graphical format that better emphasize the finding. This would be addressed by taking up the suggestions about individual data provided above. The comment "On average, applications from White PIs received slightly better scores than those from Black PIs" should be reconsidered. As the authors are well aware, the 9 point rating system introduced discontinuities around whole digit scores due to post discussion scores coalescing at the same number and panels voting within the range. Thus, "slightly" different scores, say, anything below a 20 or 30 versus those round numbers can have a dramatic impact on percentile and the probability of funding. It is best to keep such shading of the outcome to a minimum and just report that there was a difference.

Discussion, Line 275-276. It is unclear if this is referring to the difference in this study or the Ginther/Hoppe findings. The "also pertinent" suggests the latter so please clarify and cite if relevant.

Discussion, Line 324. It seems very strange to say that a 59% hit rate, or 70% miss rate is "usually". Just report the percentages without this sort of shading of the results.

Discussion 329-336: The authors are to be congratulated for including this Discussion.

Discussion Line 366: I may have missed this but it would seem imperative to re-run the analyses with the correctly identified 22% removed from the sample.

Discussion, Line 369-381: Structurally, this should be occurring right after the issue is introduced at Line 297.

Discussion, Line 385-386: This is a statement directly discordant with the primary finding that applications with white investigators score more poorly when anonymized. It should be removed.

Discussion: It is very peculiar that the manuscript has no consideration whatsoever of a recent publication by Hoppe and colleagues describing the impact of, essentially, scientific key words and how they are used by applicants of different races, on review outcome. That paper is cited only for the Black/white disparity but not for the critical new information on topic and methods.

---

## [Author Response]

Reviewer #1 (Recommendations for the authors):This is a new study but additional information would be helpful:(1) Little information is provided about the reviewers, what was their characteristics (age, gender, race, degree etc).

We did not collect any demographic information on reviewers and recorded only their prior service in the IRGs and study sections. There was concern that collecting information would have made recruitment of reviewers more difficult (i.e. that reviewers might be concerned that we would evaluate them) and we needed to recruit a large number of experienced reviewers. The demographics of these reviewers should be roughly consistent with what is published online now at CSR (see https://public.csr.nih.gov/AboutCSR/Evaluations).

(2) My understanding is that the NIH shortened the R01 application from 25 to 12 pages in 2011 with the idea that the reviewers track record (eg. biosketch) would weigh more heavily in the reviewers assessment of their ability to complete the proposed studies. In the current study the biosketch is redacted eliminating that metric. Would they have expected to see less of a difference in scores if the application were 25 pages long?

Would a longer scientific narrative lead reviewers to attend more to the science proposed and less attention to the biosketch? Perhaps so, although having served as an NIH reviewer reading both 25 page and 12-page applications, it is clear to me that the biosketches carried a lot of weight for many reviewers even when applications were longer. Reviewer 3 raises a related point about the structure of NIH review criteria (item #17, below), which we’ve addressed. We think our discussion of the effects of redacting information (e.g. line 398-408) and halo effects (lines 408-413) bear on this, at least tangentially.

(3) What was the race breakdown of the MPI's? Did presence of a White PI on an application from a Black PI impact the score?

Distribution of MPI PIs according to race: In the set of 400 applications from Black contact PI’s were 98 MPI applications. Of these, 66 had a White MPI, sometimes 1, sometimes 2. The remainder had a mix of non-White PIs. For the matched White sample of applications, 69 of the 71 MPI applications had only White MPIs. For the randomly selected applications from White PIs, 81 of 83 MPI applications had only White MPIs.

We did not do an analysis comparing Black MPI applications that included White MPIs with other applications. Because the number of such application is small, power would be low. We know that applications from White PIs scored better overall and it is not clear how this analysis help us understand the main questions of this study.

(4) The authors state that reviewers looked at 3.4 applications/reviewer. However the range was quite broad (1-29). This could introduce bias if some reviewers had large number of reviews. What was the median number of grants reviewed per reviewer?

Median number of reviews = 3. Mean number of reviews = 3.4, range 1-29. Only 60 reviewers (less than 3%) reviewed more than eight applications. When we removed data from these reviewers from the data we saw small changes in parameter estimates but no change in the patterns of significance. We now report these data, lines 188-189.

(5) Established investigators work in specific areas, an experienced reviewer will recognize the topic of the grant as most likely coming from one of these specific investigators. Redaction is not likely to help for this group of investigators. Would caps on number of awards or dollar amount to established investigators help? What percentage of the grants reviewed had PI's with 3 or more NIH awards?

We agree that highly funded scientists are likely to be better known and consequently more likely to be identified even when applications are redacted. We did not compile characteristics of the investigators who were identified. A few years ago, NIH ran analyses suggesting that the return on investment diminished with high levels of NIH funding. Policies were implemented to ensure that awards to highly funded investigators got special scrutiny before they were approved have not been strikingly successful.

(6) An issue is that the group of reviewers from 2014-2015 is not completely comparable to a reviewer from 2020-2021. Thus, the data presented may really only applies to one see specific point in time in the review process.

Caution in generalizing findings is always warranted. CSR guidance on evaluating the qualifications of reviewers has evolved over the last few years and review panels now include more assistant and associate professors and are modestly more diverse with respect to race, ethnicity, and gender than six years ago. Changes in the reviewer pool are only one reason the results obtained from this sample might not replicate now. Review policies and practices have evolved; NIH has increased attention to rigor and reproducibility in science, OER has taken more enforcement actions regarding integrity in review. However, the funding disparity that Ginther described in 2011 with 2006 data persisted, unchanged at least until last year, and recent papers (e.g. Hoppe, 2019) continue to report a differential in average scores for Black and White PIs. Therefore, we believe these findings to be highly relevant to current NIH peer review.

(7) As the authors acknowledge the dynamics of scoring and review are very different when the process involves meeting and defending your review in front of a group of your peers. Is there a way to assess whether the "quality" of the reviews was different. For example was a subset of reviews sampled by a third party and assessed as high quality?

We did not formally compare the critiques obtained for this study to those typical in CSR review. In the manuscript we emphasize differences between the review procedures of this study and true NIH review, particularly in the discussion lines 446-458 and acknowledge this is a study limitation. See our response to item 21.

Reviewer #2 (Recommendations for the authors):1) Please clarify the areas of expertise for the reviewers versus the areas of applications.

Reviewers were matched to applications by the SROs. SROs received clusters scientifically related applications. These very roughly corresponded to Integrated Review Groups at CSR, and SROs were given lists of reviewers who had reviewed those topics between October 2013 and December 2016 in the same study sections where the 1,200 applications had been originally reviewed. SROs were given, for each reviewer, names of the study sections where they had served, their areas of expertise as self-described in the NIH system or as provided by a CSR SRO who had worked with them previously, and Research, Condition, and Disease Categorization (RCDC) terms that were weighted with computed scores based on automated analysis of applications they had submitted to NIH. RCDC terms are system-generated tags applied by NIH to all incoming applications, designed to characterize its scientific content and to facilitate reporting of funding patterns. Reviewers were assigned applications to review as follows: the contract SROs first reviewed the application project narrative, abstract, specific aims, and research strategy and characterized them using key words to tag the scientific topic and critical aspects of the scientific approach. Once the key words were identified for a specific application, SROs matched them to potential reviewers’ RCDC terms and scores. In no case were reviewers assigned applications they had seen in the original reviews. A total of 6 reviewers were recruited for each application, 2 reviewers being designated as Reviewer Role 1 (best match), 2 reviewers being considered as Reviewer Role 2 (next best), and 2 reviewers as Reviewer Role 3 (still a reasonable match). We summarize these procedures in Methods, lines 169-175.

2) Please include a general breakdown of applications from varying races to the NIH.

See our response below.

3) Please include funding rates (%'s) for each race class. This really should be included in the beginning to help the readers understand the overall composition of grants versus race and their funding percentages.

We rewrote the introduction, adding information about funding rates for Hispanic and Asian PIs (the 2 largest groups of minority applicants), and provided a stronger explanation for why this study focused on Black-White differences only (lines 93-102) Our aim was to provide a broader context while keeping the intro reasonably focused. Demographic differences in patterns of application numbers, review outcomes, and funding success is a complex topic, not easily presented concisely. More importantly, we think that this information, while no doubt of interest to some, is not relevant background to the experiment at hand. We tried to strike a balance between context and focus.

4) For increased clarity about early stages, please spell out ESI in the table legends.

Done.

Reviewer #3 (Recommendations for the authors):Introduction: The opening lines jump straight into "the process" of peer review rather abruptly. This paragraph could be re-arranged slightly for a better lead-in. NIH needs to be spelled out and maybe a good entry would be a description of how much in public funds the NIH distributes for research purposes. Then the sentence at lines 43-44 should be put before the launch into bias in peer review part. This would then permit the end of the sentence on Line 38 to include mention of the interests of the US public who pay for this research, for completeness. This is not merely a matter of idle academic fluffery and the individual interests of scientist, and this should not be lost with inside-baseball framing.

We rewrote the introduction to incorporate most of the reviewer’s points (which were thoughtful and quite helpful). The intro now frames the issue more broadly, noting the importance of peer review in US funding of biomedical research, provides better justification for focusing on Black-White differences (while acknowledging other demographic disparities have been observed), eliminates the Forscher reference, and points out that NIH grant review, unlike other peer reviews, specifically calls for evaluation of the investigators and environment.

Introduction, Line 54-56. There are so many obvious flaws with the work of Forscher et al., that it is inappropriate to cite unquestioned as a negative outcome. They used laughably cartoonish stimuli, did not account for reviewer penetration of their intent and even discovered that many of the reviewers caught on to them. The present study is a considerable improvement and the only place for mentioning Forscher et al., is in drawing a contrast in how this study has improved on their glaring shortcomings.

Fair points. We’ve removed the reference.

Introduction, Line 57: Somewhere in here the authors should point out that not all peer review is the same and highlight the degree to which NIH grant review focuses explicitly, as one of five co-equal criteria, on the Investigator. This is one of the contributing reasons why available prior work may not be sufficient and further underlines the importance of their investigation. There are important differences between NIH grant review and, say, evaluation of legal summaries or resume-based job call-backs, familiar to some of the most frequently mentioned literature on this topic. This should be revisited in the Discussion around the comments being made in Line 338-341.

That’s a good point, and we’ve added it to the intro, lines 57-60, and the Discussion, lines 404-406.

Introduction, Lines 60-64: This should be considered as an alternate opening to the Introduction. Much catchier than either the comparatively narrow academic issue of peer review, or even the suggestion made above to open more broadly with the mission of the NIH.

The intro was re-written, as described above.

Introduction: While it is understood why the NIH has focused so exclusively on the disparity for Black investigators after the 2011 report, there should be some explicit recognition in the Introduction that the original Ginther found various disparities for Hispanic and Asian PIs that have not been as rigorously examined in follow-up studies such as the present one.

The intro now provides data on Asian and Hispanic funding disparities and a better explanation of why the study focuses on Black-White differences (lines 93-102).

Design: The description of reviewer tasks needs to better clarify the procedure. Were reviewers asked to provide their scores and only afterward asked all of the probing questions likely to tip them off about the true purpose of the study? Or did they know before doing their scoring that they would be asked about whether they guessed π demographics, who it was and ratings of grant writing/preparation issues.

We explain the sequence of data collection in lines 199-201. The question of what did the reviewers “know” about applicants’ race (and other demographics), and when did they know it in relation to judging the scientific merit of the application matters but is hard to answer with certainty. This was not an experiment in which perceptions of race were manipulated before the applications were scored. Rather, reviewers formed an impression of π race based on the application materials at some point in the process. We do not know when in the process the impression was formed or what it was based on. We do know many guesses were very uncertain (rated as 1 or 2 on a 5-point scale from “1. not at all certain” to “5. Completely certain”). These cautions complicate interpretation of the effects of reviewer guesses of race, and we point this out in the Discussion, lines 379-383.

Design: It is disappointing in the extreme that five of the nine SROs recruited for the study had no prior experience as an SRO. SRO reviewer selection and assignment is a hugely influential and almost entirely opaque aspect of grant review. It would seem essential in a study of this nature to include only experienced SROs. Obviously this cannot be fixed but it should be mentioned as a significant caveat in the Discussion. The points being made at lines 297-298 would be a good place- one possible reason for the unexpected outcome is that the experimental reviewers were somehow systematically different from real panel members.

The general issue raised is how differences between the experimental and actual NIH review affected the results and whether the results can be interpreted as likely to reflect patterns in real NIH review. The experiment replicated actual NIH review to the greatest extent feasible given the scale of the study. We used only experienced NIH reviewers and standard NIH criteria, critique templates, and real NIH grant applications. Reviewers for the experiment were all experienced NIH reviewers who had served on the same study sections as had originally reviewed the applications. However, there were numerous differences between the reviews done for the study and actual NIH review, which we point out in the Discussion (lines 446-458). We acknowledge that not all SROs had experience as NIH SROs and that the procedures for matching reviewers to applicants did not fully replicate those used in NIH review. Whether these differences had any effect on the degree of potential bias expressed in the reviews is unknown.

Certain findings diminish concerns that the experimental procedures greatly altered review outcomes. The overall distribution of scores obtained in the experiment closely approximates the distribution of preliminary overall impact scores seen in the actual NIH reviews of these applications. Model 2 Table 7 replicates multiple findings previously reported using data sets of scores from real NIH review, for example, that type 2 applications, and A1’s do better, and that applications from early stage investigators do worse. We agree that it is important that readers understand the differences between the experimental procedures and standard NIH review, and we point this out in the Methods (lines 190-196) and Discussion (lines 446-458).

Review Procedure: The range of reviews per-reviewer was 1 to 29. It would be useful to provide better summary statistics, i.e., include the median and Inter-quartile range for this. It is an important issue and it may be best to trim the sample a bit so that the loads are more balanced and more reflective of typical reviewer loads. These authors know perfectly well that real CSR reviewers have pressures that lead to ranking within their piles, putting closely-meritorious proposals in micro-competition. This is very likely a place where very small and implicit biases could act and this should definitely be addressed in the Discussion, the paragraph from Line 369-381 would be a good place. But this study/data set would appear to be a key opportunity to evaluate if, for example, mean scores of reviewers that only have 1-3 apps to review differ from those from reviewers who have 8-10 to review (a more typical load).

We now report that the number of reviews completed was 3.4 applications per reviewer on average (Median = 3, interquartile range = 1-5, maximum 29) (lines 188-189). Only 60 reviewers (less than 3%) reviewed more than eight applications. When we removed data from these reviewers from the data we saw small changes in parameter estimates but no change in the patterns of significance. Because the distribution is so highly skewed, we do not have sufficient sample size to perform the suggested comparison of typical load vs. study load. The point that implicit biases might emerge as reviewers rank applications within “their pile” is a good one, and we’ve added it to the discussion (lines 449-452).

The results of Pre-registered Question 1 describe the reviewer confidence on guessing race for redacted applications as "low" and to be "modestly higher" for standard applications. Yet the rather dramatic and study-relevant decrease in of identification of Black applicants from 59% to 30% (highly relevant when success rates overall at the NIH are only about 11% for Black applicants, going by the published data) is not characterized in such interpretive terms. Suggest being consistent in Results terminology. Either leave all such shading to the Discussion or include accurate shading for all Results sections.

We eliminated interpretive terms reporting that result (lines 227-243).

One important addition to the study under Q1 would be to assess scores for misses and false alarms relative to accurate identifications of π race. This would depend on there being applications on which some reviewers inaccurately identified race and other reviewers guessed correctly, but this should be summarized in any case. It would seemingly be important if, for a given application, reviewers were either all likely to mis/identify or were more randomly distributed.

The distribution of guessed race in relation to actual race showing hits and the distribution of misses is presented in Table 2, broken down according to application format. The new table gives more information and we revised the accompanying text accordingly (lines 227-243). Considering the small number of guesses per application, it is difficult to confidently identify applications that are more or less likely to have the π misidentified. However, the multivariate models presented in Table 7 address the question of how judgements of race, some accurate, some not, affect the critical test of the study hypothesis. Model 3 tests the study hypothesis using the randomly selected White applications, and found a significant Format X PI Race interaction. Model 4 adds to that variable set reviewer guess of π race, gender, institution, career stage and grantsmanship indicators. You can see that the parameter estimates for the interaction term change a little, but significance levels and pattern of findings does not. We believe it important that readers do not over-interpret the data on reviewer guesses of race, and added explanatory text in the discussion (lines 379-383).

The results of Pre-registered Question 2 describe the scoring distributions for the three groups under both standard and blinded conditions. They note the difference between standard scores for the Black and white "matched" groups despite being matched on original scores from the initial review process. This outcome indicates perhaps that there is inconsistency in the outcome of review based on race, which would seem to be critical for the NIH to understand better. Although any given application is reviewed once, given amendments and the need for multiple similar proposals to be submitted it is relevant if an applicants' race dictates higher or lower variability in scoring. It is critical to include two additional plots. First, a correlation, by group, between the original scores and the ones generated in this test under each of the standard and blinded conditions. Second, the authors should include a repeated-measures type plot where the individual applications' standard and redacted scores within this study can be visualized. This may require breaking the 400 proposals into score range bins for clarity but it would be crucial contribution to understanding what is driving effects.

The reviewer asks whether the findings indicate differential reliability of review according to race given that scores for the White matched group improved on rereview but scores for the Black group did not. We believe a more likely explanation is that the change in White matched scores represents regression to the mean. The set of applications from Black PIs represented almost the entire population of such applications. The matched applications from White PIs were drawn from a population of 26,000 applications which, on average, scored better than the Black applications. Each observed score has a true score component and an error component, presumed random. Selecting applications to match a score worse than the population mean risks selecting applications where the error terms are not normally distributed but rather are skewed negatively (making the scores worse). When they are re-reviewed, the error terms are expected to conform to a normal distribution, and thus the scores overall would improve.

The issue of what factors are associated with differences in reliability of review is interesting and important. It is a substantial issue in its own right, one that is complicated by differences between the review procedures of this study and those of standard NIH review. We think it tangential to the central point of this paper, difficult to address properly in brief form, and so would rather exclude it. Our analyses to date found higher ICCs for applications from Black PIs than for Whites, but further analyses to inform interpretation of that observation are needed.

To more specifically address the request for “a correlation, by group, between the original scores and the ones generated in this test under each of the standard and blinded conditions”, we are concerned that these correlations would not be informative. The original scores and study scores were obtained under different review conditions, as discussed in Methods and Discussion and further detailed in the supplemental materials. Given different review environments, what information is gleaned from a correlation between original and study scores? As an alternative to spaghetti plots, we report the distribution of change scores between standard and redacted conditions according to race (lines 283-285). These distributions do not show evidence of higher variability in change scores according to PI race.

**Author response table 1. sa2table1:** Distribution of change (redacted score – standard score) according to PI race.

	Min	1^st^ quarter	Median	Mean	3^rd^ quarter	Max
Black	–3.67	–0.67	0	0.04	1	3.33
White matched	–3	–0.67	0	0.16	1	3.33
White random	–3.83	–0.33	0.33	0.24	1	4.33

The results of Pre-registered Question 3 address the study's main hypothesis. This is the main point and the data should be provided in graphical format that better emphasize the finding. This would be addressed by taking up the suggestions about individual data provided above. The comment "On average, applications from White PIs received slightly better scores than those from Black PIs" should be reconsidered. As the authors are well aware, the 9 point rating system introduced discontinuities around whole digit scores due to post discussion scores coalescing at the same number and panels voting within the range. Thus, "slightly" different scores, say, anything below a 20 or 30 versus those round numbers can have a dramatic impact on percentile and the probability of funding. It is best to keep such shading of the outcome to a minimum and just report that there was a difference.

To present the results more clearly, we reformatted Figure 2 so that the effect of format on scores can be more clearly seen, and better compared between groups.

We appreciate the reviewer’s concern about shading the description of differences in a way that might be misleading. However, we believe it important to provide some sort of characterization of the difference. Effect sizes are more informative than p values, and it is effect size that we are trying to convey. We reworded the text to make it clear we are discussing effect size in a statistical sense (271-272).

Although not impossible, we think it unlikely that small effect size differences are having large effects on funding outcomes. For that to be true there would need to be many applications in close proximity to a standard NIH funding line. As shown in Figure 2, scores for these applications are widely distributed, as is typical for NIH applications. In addition, funding lines vary substantially from institute to institute (see Lauer, et. al 2021), so there is no one score that determines funding across NIH. Further, all institutes skip some highly scored applications and instead pay others, less well scored, these “select pay” decisions reflecting institute scientific priorities and portfolio considerations (see Mike Lauer’s blog). Thus, we believe that statistically small effects are in this case also likely to have relatively small real-world effects.

Discussion, Line 275-276. It is unclear if this is referring to the difference in this study or the Ginther/Hoppe findings. The "also pertinent" suggests the latter so please clarify and cite if relevant.

Done

Discussion, Line 324. It seems very strange to say that a 59% hit rate, or 70% miss rate is "usually". Just report the percentages without this sort of shading of the results.

Entire section rewritten, shadings eliminated (lines 378-390).

Discussion 329-336: The authors are to be congratulated for including this Discussion.

Thank you. The section remains in the discussion, (388-397).

Discussion Line 366: I may have missed this but it would seem imperative to re-run the analyses with the correctly identified 22% removed from the sample.

We did so, and it did not change the parameter estimates or significance levels appreciably.

Discussion, Line 369-381: Structurally, this should be occurring right after the issue is introduced at Line 297.

We agree, but decided to place it later because putting it up front gets in the way of discussing more important effects.

Discussion, Line 385-386: This is a statement directly discordant with the primary finding that applications with white investigators score more poorly when anonymized. It should be removed.

Done.

Discussion: It is very peculiar that the manuscript has no consideration whatsoever of a recent publication by Hoppe and colleagues describing the impact of, essentially, scientific key words and how they are used by applicants of different races, on review outcome. That paper is cited only for the Black/white disparity but not for the critical new information on topic and methods.

This is now discussed in the intro, lines 65-69.